# Protocol for a prospective, controlled, cross-sectional, diagnostic accuracy study to evaluate the specificity and sensitivity of ambulatory monitoring systems in the prompt detection of hypoxia and during movement

Carlos Areia [1], Sarah Vollam [1], Philippa Piper,[2] Elizabeth King,[2] Jody Ede [1], Louise Young,[1] Mauro Santos,[3] Marco A F Pimentel,[3] Cristian Roman,[3] Mirae Harford [1], Akshay Shah,[4] Owen Gustafson,[2] Matthew Rowland,[1] Lionel Tarassenko,[3] Peter J Watkinson [1]

For numbered affiliations see end of article.

**Correspondence to**
Mr Carlos Areia;
carlos.morgadoareia@ndcn.ox.ac.uk

## ABSTRACT

**Introduction** Automated continuous ambulatory monitoring may provide an alternative to intermittent manual vital signs monitoring. This has the potential to improve frequency of measurements, timely escalation of care and patient safety. However, a major barrier to the implementation of these wearable devices in the ward environment is their uncertain reliability, efficiency and data fidelity. The purpose of this study is to test performance of selected devices in a simulated clinical setting including during movement and low levels of peripheral oxygen saturation.

**Methods and analysis** This is a single centre, prospective, controlled, cross-sectional, diagnostic accuracy study to determine the specificity and sensitivity of currently available ambulatory vital signs monitoring equipment in the detection of hypoxia and the effect of movement on data acquisition. We will recruit up to 45 healthy volunteers who will attend a single study visit; starting with a movement phase and followed by the hypoxia exposure phase where we will gradually decrease saturation levels down to 80%. We will simultaneously test one chest patch, one wrist worn only and three wrist worn with finger probe devices against 'clinical standard 'and 'gold standard' references. We will measure peripheral oxygen saturations, pulse rate, heart rate and respiratory rate continuously and arterial blood gases intermittently throughout the study.

**Ethics and dissemination** This study has received ethical approval by the East of Scotland Research Ethics Service REC 2 (19/ES/0008). The results will be broadly distributed through conference presentations and peer-reviewed publications.

**Trial registration number** ISRCTN61535692 registered on 10/06/2019.

### Strengths and limitations of this study

► Controlled hypoxia exposure in a standardised environment for all participants.
► Outcome comparison with both clinical and gold standards.
► Largest study in healthy volunteers.
► Once specificity and sensitivity have been established in healthy volunteers; devices will be tested in the target hospital population.

## INTRODUCTION

Failure to recognise and act on physiological indicators of worsening acute illness in hospital wards is a prevalent problem recognised over twenty years ago.[1] Current practice involves the use of early warning scoring systems which monitor standard vital signs. These include intermittent measurements of pulse rate, respiratory rate, blood pressure, oxygen saturations and temperature. The frequency of vital signs measurements is usually guided by the clinical condition of the patient.

Intermittent measurement of these vital signs can be time consuming for healthcare professionals,[2] and therefore the desired frequency of observations is often not achieved.[3] Infrequent measurement of vital signs may also miss clinical deteriorations between these measurements.[4] Thus, more sustainable, accurate and less time-consuming monitoring methods would be highly desirable.

Wearable ambulatory monitors (AM) may provide an alternative to intermittent vital signs monitoring by enabling the continuous monitoring of vital signs parameters. In addition to reducing the burden of intermittent

BMJ

measurement of vital signs on staff, continuous monitoring has the potential to facilitate earlier detection of deranged physiological parameters.[5] A major barrier to the clinical implementation of these wearable devices is their uncertain reliability, efficiency and data fidelity.[6] In particular, the effect of motion on its accuracy is under-investigated. Recent work by Louie et al[7] tested four non-ambulatory pulse oximeters and found that motion impaired performance throughout a clinically relevant range of measurements. Less accuracy was reported at lower arterial oxygen saturations ($SaO^2$) which is undesirable in clinical practice.[7]

This study is part of the Virtual High Dependency Unit (vHDU) project, a collaboration between the Institute of Biomedical Engineering and clinicians from the Nuffield Department of Clinical Neurosciences at the University of Oxford. This is a phased project aiming to refine and integrate ambulatory monitoring systems for use in clinical practice. Previous phases have tested device wearability and in situ testing on hospital wards. The purpose of this study is to test the performance of selected devices in a simulated clinical setting which will involve participant movement and inducing low peripheral oxygen saturations.

## METHODS

This protocol follows the Standard Protocol Items: Recommendations for Interventional Trials reporting guidelines.[8]

### Study objectives

Primary objective: To determine the specificity and sensitivity of currently available ambulatory vital signs monitoring equipment for the detection of hypoxia.

Secondary objective: To determine the effect of movement on data acquisition by currently available ambulatory vital signs monitoring equipment.

These objectives will be assessed by comparing continuous peripheral oxygen saturations, pulse rate, heart rate and respiratory rate data from each AM with arterial blood oxygen saturation measured through arterial blood sampling, pulse rate derived from the arterial blood signal, heart rate derived from standard care 3-lead ECG, and respiratory rate derived from capnography and manual counting.

### Study design

Prospective, observational cross-sectional cohort study. Vital signs parameters from study devices will be compared with 'gold standard' and 'clinical standard' measurements.

### Sample size

Our sample size calculation is based on the ISO 80601-2-61:2019 guideline for pulse-oximetry equipment accuracy testing. This requires at least 200 data points balanced across each decadal range (70%–80%, 80%–90%,

90%–100%) of the $SaO^2$ range 70%–100%, from at least 10 subjects. Approximately 30 full data sets will be required, to yield sufficient data points for the primary and secondary outcomes; therefore, up to 45 healthy adult subjects who meet the inclusion criteria will participate in the study. For the broadest application to the largest group of participants, the subjects should vary in their physical characteristics to the greatest extent possible.

### Recruitment

Up to 45 healthy volunteers will be recruited, with adverts placed in appropriate target locations such as college common spaces and university buildings. The adverts will contain a description of the study and the number and email contacts of one of the members of the research team.

Inclusion criteria for the study are: willing and able to give informed consent for participation in the study; men and women aged 18 or over; and in generally good health.

Exclusion criteria are: allergies to adhesive dressings (such as bio-occlusive dressings or micropore) or local anaesthetic (eg, lidocaine); intracardiac device (eg, permanent pacemaker) or previous wrist arterial line; epilepsy; angina, congenital heart disease or history of severe cardiopulmonary disease; history of anaemia (reported in the prescreening telephone call), haemoglobinopathy or haemoglobin below $100\,g/L$ on first test; resting hypoxaemia (SpO2 <94%) or significant cardiopulmonary disease rendering exposure to alveolar hypoxia unsafe, as determined by the research physician; pregnancy or breast feeding; clotting disorders and use of antiplatelet or anticoagulant medication (such as aspirin); and claustrophobia precluding spell in the hypoxic exposure.

### Study procedures
#### Initial contact

Healthy volunteers will contact the research team via telephone/email to express their interest in the study. The research team will provide further information including the Participant Information Sheet (PIS). If volunteers wish to proceed, a telephone appointment will be arranged to complete a brief prescreening assessment with a research nurse/physiotherapist (supported by a senior anaesthetist).

#### Pre-screening assessment

During the prescreening telephone appointment, the study will be discussed further and general screening questions will be asked, to confirm eligibility. Questions will be encouraged to ensure the potential participant understands the study. If the potential participant agrees to take part, an appointment for the hypoxia exposure visit will be agreed.

### Study visit
#### Screening assessment

The screening assessment will be completed by an appropriately qualified, medically trained member of the

| Table 1 | Device combinations | | |
|---|---|---|---|
| Devices placement | Combination 1 (n=10) | Combination 2 (n=10) | Combination 3 (n=10) |
| Chest | VitalPatch® | VitalPatch® | VitalPatch® |
| Wrist | Wavelet | Wavelet | Wavelet |
| 1st (thumb) | CheckMe™ O2+ | CheckMe™ O2+ | CheckMe™ O2+ |
| 2nd (index finger) | Philips monitor (MX450) | WristOX2 3150 BLE | AP-20 |
| 3rd (middle finger) | AP-20 | Philips monitor (MX450) | WristOX2 3150 BLE |
| 4th (ring finger) | WristOX2 3150 BLE | AP-20 | Philips monitor (MX450) |

research team, who will confirm eligibility for the hypoxic exposure phase. This will include a urinary pregnancy test for all female participants of childbearing potential. Pregnancy is an exclusion criteria for the study as the effects of hypoxia on pregnancy are unknown.[9]

### Arterial line insertion
After confirmation of eligibility, an arterial line will be inserted into the non-dominant radial artery of each participant on the day of the study visit under local anaesthesia.

### Initial blood gas sampling
The first arterial blood gas (ABG) measurement will be assessed by the anaesthetist to confirm haemoglobin concentration ≥100 g/L. If the haemoglobin is below this level, the participant will be withdrawn from the study, the arterial line removed and the participant advised to discuss this finding with their general practitioner (as stated in the PIS).

### Placement of devices
Participants will wear several ambulatory monitoring devices (AMD) which may include a chest worn patch (VitalPatch®), a purely wrist worn device (Wavelet) and up to three wrist-worn devices with finger probe (CheckMe™ O2+, AP-20 and WristOX2 3150 BLE). The AMD detect various combinations of pulse oximetry, pulse rate, heart rate and respiratory rate (please refer to online supplementary appendix 1 for device details). Participants will be asked to wear a maximum of five study devices during the hypoxia exposure visit (figure 1).

### Finger probe position randomisation
Up to three study device finger probes will be worn, in addition to the 'clinical standard' reference bedside monitor finger probe. The CheckMe™ O2+ will always be worn on the thumb as per manufacturer recommendation.[10] To ensure parity of testing, the position of the other three finger probes on the second, third and fourth fingers will be randomised (using https://www.random.org/) per study visit day, ensuring an even distribution of placement, as per table 1).

### Stage 1: movement phase
During the movement phase, participants will be seated in a chair and asked to complete a series of consecutive standardised movements, as detailed in table 2. An ABG and manual (counted) respiratory rate will be taken at the end of each movement.

### Stage 2: hypoxia exposure phase
Participants will move to a bed and lie comfortably in a semi-recumbent, supine position (figure 2). A tight-fitting silicone face mask will be placed and connected to a hypoxicator unit (Everest Summit Hypoxic Generator, www.altitudecentre.com). If required, additional 7% oxygen in nitrogen from a cylinder will be entrained into the hypoxicator circuit to ensure tight control of fraction of inspired oxygen ($FiO_2$) provided to the participant.[11] Inhaled $FiO_2$ will be monitored by an in-line gas analyser and end-tidal carbon dioxide ($etCO_2$) will be also recorded via capnography using the Philips monitor MX450 (www.philips.co.uk).

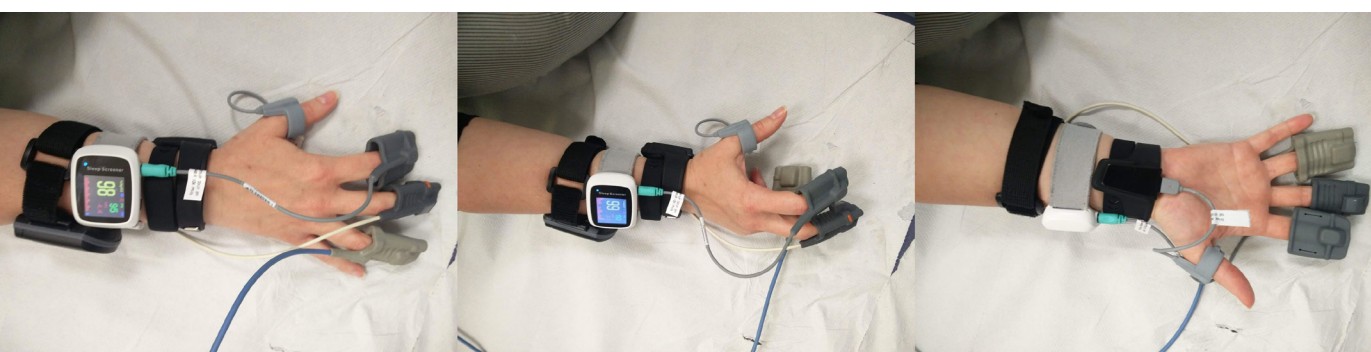

**Figure 1** Devices placement example (dominant hand, combination 3).

**Table 2** Standardised movements to be tested

| Movement | Task |
|---|---|
| Standing from chair using arms to push up/sit down | 20x repetitions. |
| Tapping | Volunteer to tap a surface with all test side fingers simultaneously at the speed of a metronome set at 100 bpm for 2 min. |
| Rubbing | Volunteer to complete a sideways rubbing movement with all test side fingers simultaneously at the speed of a metronome set at 100 bpm for 2 min. |
| Drinking from plastic cup | 20 x lift/drink/put down. |
| Turning page | 50x page turns. |
| Using tablet | As per protocoled instructions (online supplementary file 2). |

During the hypoxia exposure phase, oxygen saturations from the 'clinical standard' Philips monitor will guide the titration of the hypoxicator. Seven per cent oxygen in nitrogen will be used to further lower $FiO_2$ if required. An ABG will be sampled when the participant reaches and remains stable at each pre-specified target peripheral oxygen saturation level (95%, 90%, 87%, 85%, 83%, 80%). We specified these saturations to allow assessment for our use case of prompt detection of hypoxia in normal adult patients in a ward environment, including multiple assessments within the 83%–95% range, and one assessment at the top end of the 70%–80% range, considered severe hypoxia.

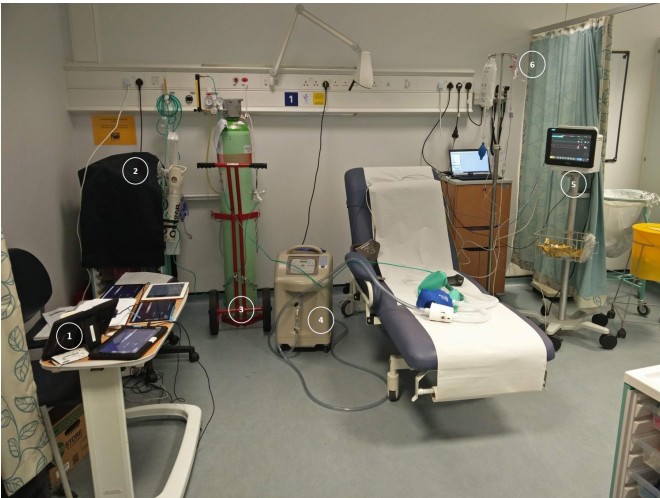

**Figure 2** Hypoxia study day set-up. Legend: 1: tablets linked with AMD devices (4 Samsung TAB A, each linked with one AMD: AP-20, WristOX2 3150 BLE, CheckMe™ O2 and VitalPatch®. 1 iPad four connected to the wavelet). 2: resuscitation trolley and oxygen. 3%–7% oxygen in nitrogen cylinder. 4: hypoxicator apparatus. 5: Philips monitor (model MX450) connected to laptop (IX trend software). 6: drip stand with the arterial line pressure bag.

### Blood sampling
Up to 15 ABG samples will be taken. Samples will be discarded at the end of the laboratory session into clinical waste and no blood will be retained by the study.

At the end of the study visit, the arterial line will be removed and firm pressure applied to the site until haemostasis is achieved. A sterile dressing will be applied and advice given to the participant on action to take if any bleeding occurs.

### Facilities and research staff
#### Facilities
All study visits will occur in the Cardiovascular Clinical Research Facility, Level 1 Oxford Heart Centre, John Radcliffe Hospital, Headley Way, Headington, Oxford, OX3 9DU.

#### Roles and responsibilities
Each study visit day will be staffed by one senior anaesthetist, four clinical researchers and one engineer. Roles during the study visit are defined in table 3.

### DATA COLLECTION AND MANAGEMENT
#### Devices
To ensure correct time-stamping, Researcher 1 will verify all devices, tablets and laptops are connected to the same network. The time and date will be set to Greenwich Mean Time Zone (GMT) or British Summer Time as appropriate. The time will be verified to be within a tolerance of ±2 s and documented in the case report form (CRF):
1. AMDs
   a. Vital Connec, VitalPatch®: Single-use (120 hours), adhesive, wireless, waterproof patch that measures heart rate and respiratory rate via a single-lead ECG. Other parameters include three axis motion sensor and skin-temperature sensors.
   b. Viatom Technology, CheckMe™ O2: Wrist-worn wireless pulse oximeter, measuring pulse rate and percentage Oxygen saturation (SpO2) via transmittance photoplethysmography (PPG) using a ring-style sensor. Other parameters include a motion sensor.
   c. Wavelet Health, USA, Wavelet wristband: Wireless wrist-worn pulse oximeter using reflectance PPG to measure pulse rate and percentage Oxygen saturation (SpO2). Other parameters include 3-axis motion sensor and gyroscope.
   d. Shenzhen Creative Industry, AP-20: Wrist-worn wireless pulse oximeter, measuring pulse rate and percentage Oxygen (SpO2) via transmittance PPG using a finger-tip style sensor. Other parameters include estimation of respiratory rate using the air-flow signal collected from a supplied nasal cannula (attached to an airflow sensor in the device); and 3-axis motion sensor.
   e. Nonin Medical, WristOX2 3150 BLE: Wrist-worn wireless pulse oximeter, measuring pulse rate and

**Table 3** Research team roles

| Professional | Role in study | Description of responsibilities |
|---|---|---|
| Senior anaesthetist | Medical cover | ► Conduct medical screening.<br>► Ensure participant safety throughout the study.<br>► Inserting/removing radial arterial line.<br>► Operating the hypoxicator equipment.<br>► Taking ABG samples from arterial line. |
| Researcher 1 | Devices and time-stamping | ► Ensure correct positioning of all involved devices.<br>► Ensure data is being recorded from all monitors.<br>► Time-stamping of study activities and ABGs.<br>► Troubleshoot any device-related issues throughout. |
| Researcher 2 | ABG processing | ► Collect and process the ABG.<br>► Identify ABG report with correct activity (eg, tapping, tablet, 95%).<br>► Discard the blood sample. |
| Researcher 3 | Participant activities and instructions | ► Explain activities to participants.<br>► Giving instructions and guide participants through movement phase activities.<br>► Respiratory rate manual count at ABG time points.<br>► $FiO_2$ manual record at ABG time points in the hypoxia phase. |
| Researcher 4 | Support/backup | ► Manually record the time ABGs are drawn.<br>► Complement/assist any required activities.<br>► Responsible for oversight and detection of any suboptimal activities/conditions. |
| Engineer | Data monitoring | ► Monitors procedures and real time data quality.<br>► Double checks devices.<br>► Ensures reliable data acquisition throughout. |

ABG, arterial blood gas; $FiO_2$, fraction of inspired oxygen.

percentage of Oxygen (SpO2) via transmittance PPG with finger-tip style sensor.

2. Clinical standard:
   a. Philips Monitor MX450 and extraction software (ix-Trend 2.1):
3. Gold standard:
   a. ABGs: The assigned person will manually record the GMT time when each ABG is taken. The ABG processing time will also be recorded as part of the automatic report. The ABGs will be analysed using a Radiometer ABL90 Flex blood gas analyser.

An electronic system, developed in-house, comprising a vital signs data collection application (app) running on Android tablets, and a web-application (administrated by the research group) will allow:

► the registration of the participant study number, centralised in the web-application, via the app.
► the collection of data from one patch (VitalPatch®) and one pulse-oximeter (AP-20, CheckMeO2™ and WristOX2 3150), via Bluetooth Low-energy, and their storage into files in a tablet.
► the upload of the files from the tablet to the web-application server (via HTTPS) within 24 hours of the end of each session.
► the electronic recording of the time-stamping of the activities in each phase of the study session (by researcher 1), that is:

– **Movement phase:** Normoxia/Sit to Stand/Tapping/Rubbing/Drinking/Turning/Tablet.
– **Hypoxia phase:** 95%/90%/87%/85%/83%/80% SpO2 levels.

A total of 3 tablets will be used to collect data from the VitalPatch®, CheckMe™ O2, WristOX2 3150 BLE and the AP-20. Wavelet Health's electronic system will be used to collect data for the Wavelet device.

## Collected data

The following data will be collected for each participant:

► Demographic data: including age, sex, height, weight, skin type (Fitzpatrick scale), baseline heart rate and $SaO_2$ at start of test (using gold standard ABG measurements).
► For oxygen saturation, sampled at normoxia and each level of induced hypoxia:
   – **Gold standard reference:** ABGs (intermittent samples).
   – **Clinical standard reference:** Standard care pulse oximeter (continuous data).
   – **Devices under test**: Up to four pulse oximeters (continuous data).
► For pulse rate, sampled at normoxia and each level of induced hypoxia:
   – **Gold standard reference:** Arterial line trace (continuous data).

- – **Clinical standard reference:** Standard care pulse oximeter (continuous data).
- – **Devices under test**: Up to four pulse oximeters (continuous data).
► For heart rate, sampled at normoxia and each level of induced hypoxia:
- – **Gold standard reference and clinical standard reference**: Standard care 3-lead ECG (continuous data).
- – **Devices under test:** Chest patch (continuous data).
► For respiratory rate, sampled at normoxia and each level of induced hypoxia:
- – **Gold standard reference**: Capnography (continuous data).
- – **Clinical standard reference**: Manual respiratory rate per minute counting (intermittent samples, done at the same time as the ABG sampling).
- – **Devices under test:** Chest patch (continuous data).

### Safety testing and calibration
Philips MX450, ABG machines, hypoxicator, all tablets and chargers were subjected to clinical safety testing by either the Department of Engineering Science, University of Oxford or by the Clinical Engineering team at the Oxford University Hospitals NHS Foundation Trust (OUHFT). ABG analysers are maintained and calibrated by the Clinical Measurements team at the OUHFT.

### Data quality and completeness
Manually entered data (eg, ABG data, RR count, etc) will be subject to a 10% data validation check.

To ensure correct time-stamping, all ABG collection times will be recorded both using the vHDU app and manually recorded time (hh:mm:ss). Each participant AMD, gold and clinical standard, and time-stamp data will be plotted and audited visually up to 1 week after participation to assess data completeness. Each participant dataset will be deemed complete if there are test device data to answer either the primary or secondary objective, including both gold standard and clinical standard reference data.

### ANALYSIS
For continuous data we will sample by two methods: (1) simultaneous single data points, (corresponding to the time of ABG sampling where relevant) and; (2) by selecting sampling windows of 5–30 s and comparing data for each device. Data points will be recorded across device time-stamps to ensure accuracy of comparisons.

In accordance with the international standard of pulse oximeter equipment validation (ISO 80601-2-61:2019), the accuracy of the SpO2 measurement will be stated in terms of the root-mean-square (rms) difference between measured values (devices under test) (SpO2i) and reference values (gold standard arterial line and clinical standard) (SRi), as given by:

$$A_{rms} = \sqrt{\frac{\sum_{i=1}^{n}\left(SpO_{2i}-S_{Ri}\right)^2}{n}}$$

We will also compute the bias between gold standard, clinical standard and each device under test:

$$B = \frac{\sum_{i=1}^{n}\left(SpO_{2i}-S_{Ri}\right)}{n}$$

and the precision:

$$s_{res} = \sqrt{\frac{\sum_{i=1}^{n}\left(SpO_{2i}-SpO_{2fit,i}\right)^2}{(n-2)}}$$

where SpO2fit is the value of the fitted curve corresponding to the 'i'th reference value. Simple statistics about the difference among measurements (mean, SD, percentiles, Bland-Altman plots) will be provided for the all the devices under analysis. Identical statistical methods will be applied to assess the agreement between the estimation of the (i) pulse rate from the pulse oximeters, and of the (ii) heart rate and (iii) respiratory rate from the patch, and the corresponding reference measurements.

The performance of each pulse oximeter in detecting hypoxaemia (at each level ≤90%) will be assessed by reporting the optimal sensitivity and specificity pair, identified via Receiver-Operating Characteristic curves.[7 12]

Descriptive statistics will be also computed per participant, per skin type and with and without movement artefacts.

### Outcome analysis
We will analyse the outcomes on all the participants from whom we collected the data. The accuracy will be compared with the respective reference pulse oximeter. Where paired readings exist (device with clinical or gold standard), they will be included in the analysis. Device readings with no paired clinical or gold standard will be excluded from the analysis.

Primary outcome measure: Sensitivity and specificity for detecting hypoxia at each level ≤90%): The analysis plan detailed above will provide data on correlation of each device with 'gold standard' arterial measurements, and their accuracy for the detection of hypoxaemia: The output of this analysis will allow selection of the devices which correlate most closely to the 'gold standard' measurements, and provide the highest performance in the detection of hypoxaemia.

Secondary outcome measure: Correlation of device outputs, that is, HR, RR, PR and SpO2, with ECG derived HR, capnography derived RR, arterial blood pulse rate and pulse oximetry, respectively, during movement:

The analysis plan detailed above will provide data on correlation of each device with 'gold standard' arterial measurements during protocolised movement tests. The output of this analysis will allow selection of the devices which correlate most closely to the 'gold standard' measurements.

### Ethics and dissemination
The results will be broadly distributed through conference presentations and peer-reviewed publications.

## Safety reporting

A serious adverse event (SAE) occurring to a participant should be reported to the Research Ethics Committee that gave a favourable opinion of the study where in the opinion of the chief investigator the event was 'related' (resulted from administration of any of the research procedures) and 'unexpected' in relation to those procedures. Reports of related and unexpected SAEs should be submitted within 15 working days of the chief investigator becoming aware of the event, using the Health Regulatory Agency report of SAE form. The research team will also request participant's permission to contact the respective general practitioner (GP) and report the SAE through the appropriate channels.

## Informed consent and participant withdrawals

Informed consent will be obtained by the lead researcher or a member of the research team (usually a research nurse/physiotherapist) at the start of the study visit (online supplementary file 3). All those obtaining consent will have received informed consent training as well as Good Clinical Practice training. Each participant has the right to withdraw from the study at any time, without giving a reason and without affecting their career or quality of their future care. If they wish to withdraw from the study, we will offer to destroy all gathered information. This will be possible up until the point where we de-identify participants' data.

## Data recording and pseudonymisation

All vital signs data will be collected as per each AMD (online supplementary appendix 1). Data derived from these devices will be limited to vital signs measurements and associated waveforms. These will be downloaded from the device to a secure, password-protected database. No personal identifiable information will be associated with these data. All data held will be associated with a de-identifiable participant number. Where data are uploaded initially to a cloud server (the Wavelet Health wristband), data will be subsequently downloaded by research staff. The download of data does not remove the de-identifiable data from the cloud storage. Access rights to the cloud data will be as per Cloud Privacy licence. Participants will be explicitly advised as to the storage of de-identifiable vital signs data and that this data may be kept within the storage facility indefinitely. This will be made clear to participants prior to consent. Other AMD will record data directly to internal devices from which the data can be retrieved and deleted.

Linkage between pseudonym and identifying information will be held in one place, a password-protected database on a networked secure server held by the University of Oxford. Access to this database will be limited to research nurses/allied health professionals only and will be destroyed at the end of the study, once all data has been verified. A spreadsheet will be maintained of de-identifiable participant baseline data, such as date of participation. No identifiable data will be held on this spreadsheet. This data will be entered and validated by the study researchers.

Any paper correspondence (such as CRFs and CFs) will be kept in the Kadoorie Centre in an established research area, behind two access-controlled doors and in locked filing cabinets. All documentation will be archived at the end of the project and retained for 5 years at the off-site secure archive facility (Re-Store) based at Upper Heyford.

## Cloud storage

As these are commercially available systems, de-identifiable data, with no personal identifiers may be transferred to Cloud storage. Where this is the case, this will be discussed with participants before connecting the equipment prior to consent. Data may remain on the storage system even when downloaded by the research team. Access to storage data is as per Cloud licensing agreement. Participants will be explicitly advised as to the storage of de-identifiable vital signs data and that this data may be kept within the storage facility indefinitely. Other AMD will record data directly to internal devices from which the data can be retrieved and deleted. For one device, the Wavelet Health wristband, de-identified data are transmitted to the device manufacturer's cloud-based system before we are able to access and download these data. There is no alternative to this method of transmission for this device.

## Participant compensation

Participants will be reimbursed for travel costs incurred, plus appropriate payment in recognition for their time contribution to the study. They will receive vouchers to the monetary value of £20 for completing the pre-screening telephone interview, and then if willing and eligible to participate, £80 for the complete hypoxia exposure visit making a total of £100 for those who complete the study.

## Patient and public involvement

This study is part of the vHDU project. During phase 4 (commenced), we will develop a patient and public involvement group for ongoing support and feedback. We have attended a number of local public engagement events, where members of the public showed genuine interest in the advances of wearable monitors and were engaged in our vision of a wireless hospital in the future.

**Author affiliations**

[1]Critical Care Research Group, Nuffield Department of Clinical Neurosciences, University of Oxford, Oxford, UK
[2]Adult Intensive Care Unit, Oxford University Hospitals NHS Foundation Trust, Oxford, UK
[3]Institute of Biomedical Engineering, Department of Engineering Science, University of Oxford, Oxford, UK
[4]Radcliffe Department of Medicine, University of Oxford, Oxford, UK

**Contributors** Authorship is determined in accordance with the ICMJE guidelines: CA, SV, PP, LT and PW drafted the initial protocol. CA, SV, EK, JE, LY, OG, CR, MR, MP and MS will conduct the study procedures and data acquisition. AS and MH reviewed protocol and will provide medical cover in the study days. CA drafted the

manuscript and all authors reviewed and approved it.The funders have had no role in the study protocol design or the preparation of this manuscript, and will have no role in the collection, management, analysis and interpretation of the data or the writing of the final report.

**Funding**  This study/project is funded by the NIHR Biomedical Research Centre, Oxford. PW and LT are supported by the NIHR Biomedical Research Centre, Oxford.

**Competing interests**  PW and LT report significant grants from the National Institute of Health Research (NIHR), UK and the NIHR Biomedical Research Centre, Oxford, during the conduct of the study. PW and LT report modest grants and personal fees from Sensyne Health, outside the submitted work. PW and LT work part-time for Sensyne Health and hold shares in the company.

**Patient consent for publication**  Not required.

**Ethics approval**  This study has received ethical approval by the East of Scotland Research Ethics Service REC 2 (19/ES/0008).

**Provenance and peer review**  Not commissioned; externally peer reviewed.

**Data availability statement**  Data are available upon reasonable request. Deidentified participant data may be available in the end of the study upon reasonable requests.

**ORCID iDs**
Carlos Areia http://orcid.org/0000-0002-4668-7069
Sarah Vollam http://orcid.org/0000-0003-2835-6271
Jody Ede http://orcid.org/0000-0001-7289-6991
Mirae Harford http://orcid.org/0000-0003-2851-1577
Peter J Watkinson http://orcid.org/0000-0003-1023-3927

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
