## [Reviewer comments · BMJ Open]

ARTICLE DETAILS

TITLE (PROVISIONAL)	Protocol for a prospective, controlled, cross-sectional, diagnostic accuracy study to evaluate the specificity and sensitivity of ambulatory monitoring systems in the prompt detection of hypoxia and during movement.
AUTHORS	Areia, Carlos; Vollam, Sarah; Piper, Philippa; King, Elizabeth; Ede, Jody; Young, Louise; Santos, Mauro; Pimentel, Marco; Roman, Cristian; Harford, Mirae; Shah, Akshay; Gustafson, Owen; Rowland, Matthew; Tarassenko, Lionel; Watkinson, Peter

VERSION 1 - REVIEW

REVIEWER	Mirela Prgomet Macquarie University, Australia
REVIEW RETURNED	02-Dec-2019

GENERAL COMMENTS	This manuscript describes the protocol aimed at evaluating the diagnostic accuracy (specificity and sensitivity) of ambulatory monitoring systems for the prompt detection of hypoxia and during movement. I found this to be an excellent study protocol - it is clear, detailed and well described. Thank you for the opportunity to review such a carefully considered and well written piece of work.
---

REVIEWER	Grzegorz Bilo University of Milano-Bicocca, Milan, Italy
REVIEW RETURNED	06-Dec-2019

GENERAL COMMENTS	Areia et al. present a protocol for evaluating the diagnostic value of ambulatory systems for vital signs monitoring focusing on measurements during movement and aimed to detect low oxygen saturation. The authors propose to study 45 health volunteers in a single centre study. The study is currently ongoing, according to the authors' declaration. The protocol describes the study procedures in detail and clearly (with some exceptions reported below). The study has the potential to influence current practice standards by supporting the use of wearable devices in inpatients vital signs monitoring. To achieve this the diagnostic accuracy of these devices must be shown. I have a few comments:
--

	1. What is the rationale for including VitalPatch in this study – this device is very different from the others in terms of collected signals, in particular it does not measure oxygen saturation. The latter is relevant given that the protocol was designed based on ISO guideline for validating oximetry devices. 2. Table 1: 10 participants per each of the three random combinations are foreseen but the total number of participants is 45. Please clarify. 3. Table 2: Please clarify which body part will be used to performed “tapping” and “rubbing” (if finger then which). 4. Hypoxia exposure phase – please explain in more detail the procedure: what is the FiO₂ generated by the Everest device; what is the planned timing of this sequence (time needed to achieve given SpO₂ level and time required to define it as stable).
--	--

VERSION 1 – AUTHOR RESPONSE

Reviewer: 1

This manuscript describes the protocol aimed at evaluating the diagnostic accuracy (specificity and sensitivity) of ambulatory monitoring systems for the prompt detection of hypoxia and during movement. I found this to be an excellent study protocol - it is clear, detailed and well described. Thank you for the opportunity to review such a carefully considered and well written piece of work.

Areia et al. present a protocol for evaluating the diagnostic value of ambulatory systems for vital signs monitoring focusing on measurements during movement and aimed to detect low oxygen saturation. The authors propose to study 45 health volunteers in a single centre study. The study is currently ongoing, according to the authors' declaration. The protocol describes the study procedures in detail and clearly (with some exceptions reported below). The study has the potential to influence current practice standards by supporting the use of wearable devices in inpatients vital signs monitoring. To achieve this the diagnostic accuracy of these devices must be shown. I have a few comments:

We would like to thank Reviewer 1 for taking the time to review our protocol and for such an encouraging statement on our work.

Reviewer: 2

The team would like to thank Reviewer 2 for the time and valuable comments to this protocol. You may find the answers below and any changes in the re-submitted protocol.

1. What is the rationale for including VitalPatch in this study – this device is very different from the others in terms of collected signals, in particular it does not measure oxygen saturation. The latter is relevant given that the protocol was designed based on ISO guideline for validating oximetry devices.

Many thanks for your question.

Our definition of Ambulatory Monitoring System (AMS) in the protocol is a combination of devices that estimate SpO₂, HR and RR. Given that, within our tested AMS, only the pulse oximeters are capable of SPO₂ estimation, it's easy to conclude that those devices have more weight in computing the outcomes related with the hypoxia phase of the study. However, stating that the protocol is designed

based on the ISO for validating oximetry devices is an over-simplification. We note that the protocol has two distinct phases (i.e the “Activity and Hypoxia phases”), and measures two main outcomes:

“Primary objective: To determine the specificity and sensitivity of currently available ambulatory vital signs monitoring equipment for the detection of hypoxia.” – This is done through AUROC analysis after defining the hypoxia and normoxia ranges (and this methodology is not reflected in the said ISO, and was derived from different literature).

“Secondary objective: To determine the effect of movement on data acquisition by currently available ambulatory vital signs monitoring equipment.” – this is done by comparing the agreement of the ambulatory monitoring system (AMS) estimates (HR/RR/SpO₂) with the matching gold standard estimates. The measurements of “agreement” in the ISO 80601-2-61:2019 are not only adequate to test the pulse oximeters estimates, but can also be reused for the vital-sign estimates produced by the VitalPatch.

In responding to our secondary outcome “accuracy of the AMS during the activity phase”, we found that the VitalPatch is the only device with FDA approval, and that passed our internal wearability tests, capable of RR estimation. We therefore included it to understand, primarily, the accuracy of both its HR and RR estimation during the movement phase. Alongside, we will analyse the potential changes in HR and RR during hypoxia phase and report the agreement for HR and RR in that phase as well.

2. Table 1: 10 participants per each of the three random combinations are foreseen but the total number of participants is 45. Please clarify.
--

Many thanks for your comment.

We decided to recruit up to 45 participants to ensure we would obtain at least 30 complete datasets for the primary and secondary outcome measures. As expected, once we started recruitment, some participants became ineligible after the first arterial blood gas (haemoglobin below 100 g/l on first test, as described in our exclusion criteria). For others, we did not obtain a complete dataset (eg. unable to make hypoxic, technical difficulties with devices, etc...).

These challenges were expected from the beginning so we have decided to recruit over target to ensure at least 30 complete datasets. This information has been added to the study protocol under the “sample size” section.

3. Table 2: Please clarify which body part will be used to performed “tapping” and “rubbing” (if finger then which).
--

Many thanks for your comment.

Participants will use the hand where the devices are placed (ideally in the dominant hand) and finger allocation will be as per the combination number outlined in Table 1, randomised on the study day. Regarding the movements:

Tapping – Participants will tap all fingers simultaneously on the table at the speed of the metronome (100 beats per minute) for a total of 2 minutes

Rubbing - Participants will rub side to side all fingers simultaneously on the table (using wrist ulnar/radial deviation movements) at the speed of the metronome (100 beats per minute) for a total of 2 minutes.

Further clarification added to protocol.

4. Hypoxia exposure phase – please explain in more detail the procedure: what is the FiO2 generated by the Everest device; what is the planned timing of this sequence (time needed to achieve given SpO2 level and time required to define it as stable).

Many thanks for your comment. The Everest Summit Hypoxic Generator may provide an oxygen level as low as 12.7% which is equivalent to about 4000m. Device and titration is similar to previously published work (Rowland et al., 2017) and inhaled FiO2 will be monitored by an in-line gas analyser.

The FiO2 generated by the hypoxicator can be found in the table below:

Approximate Hypoxicator setting	Effective Inspired Oxygen %	Effective Height (Feet)	Effective Height (meters)	Partial pressure Oxygen (mmHg)	Comments
	20.90	0	0	158.8	
	20.70	260	80	157.3	
0.5	20.50	530	160	155.8	
	20.30	800	240	154.3	
1		870	265		somewhere inbetween
1.5	20.10	1070	330	152.8	
2	19.90	1350	410	151.2	
	19.70	1620	490	149.7	
	19.50	1900	580	148.2	
	19.30	2180	670	146.7	
2.5	19.10	2470	750	145.2	
	18.90	2750	840	143.6	
3		2900	884		somewhere inbetween
	18.70	3040	930	142.1	
3.5	18.50	3330	1020	140.6	
	18.30	3620	1100	139.1	
4	18.10	3920	1200	137.6	
	17.90	4220	1290	136.0	
	17.70	4520	1380	134.5	
	17.50	4830	1470	133.0	
5		5000	1524		somewhere inbetween
	17.30	5130	1560	131.5	
	17.10	5440	1660	130.0	
5.5		5600	1707		somewhere inbetween
	16.90	5760	1750	128.4	
6	16.70	6070	1850	126.9	
	16.50	6390	1950	125.4	
	16.30	6710	2050	123.9	
6.5	16.10	7040	2140	122.4	
7	15.90	7370	2250	120.8	
	15.70	7700	2350	119.3	
	15.50	8030	2450	117.8	
7.5	15.30	8370	2550	116.3	
	15.10	8710	2660	114.8	
8	14.90	9060	2760	113.2	
	14.70	9410	2870	111.7	
	14.50	9770	2980	110.2	
8.5	14.30	10120	3090	108.7	
	14.10	10480	3200	107.2	
	13.90	10850	3310	105.6	
9	13.70	11220	3420	104.1	
9.5	13.50	11600	3530	102.6	
	13.30	11980	3650	101.1	
10	13.10	12360	3770	99.6	
10.5	12.90	12750	3890	98.0	
12	12.70	13140	4010	96.5	11-12 clustered

As mentioned in the protocol, in the hypoxia phase of the study we will stabilise participants in the specified target peripheral oxygen saturation level (95%, 90%, 87%, 85%, 83%, 80%). In order to achieve SpO2 stability at the target levels, FiO2 will be carefully titrated and SpO2 stabilised between

45-60 seconds before taking the arterial blood gas. There are no pre specified timings for this due to inter-individual variability on the time taken to reach the desired SpO2.

Reference

Rowland, M. J. et al. (2017) 'Calcium channel blockade with nimodipine reverses MRI evidence of cerebral oedema following acute hypoxia', Journal of Cerebral Blood Flow & Metabolism, p. 0271678X1772662. doi: 10.1177/0271678X17726624.

VERSION 2 – REVIEW

REVIEWER	Grzegorz Bilo University of Milano-Bicocca, Milan, Italy
REVIEW RETURNED	16-Dec-2019

GENERAL COMMENTS	The Authors have provided detailed and satisfactory replies to my previous comments and the manuscript has been modified accordingly. I have no further comments.
---